# Supervised Contrastive Learning with Angular Margin for the Detection and Grading of Diabetic Retinopathy

**DOI:** 10.3390/diagnostics13142389

**Published:** 2023-07-17

**Authors:** Dongsheng Zhu, Aiming Ge, Xindi Chen, Qiuyang Wang, Jiangbo Wu, Shuo Liu

**Affiliations:** 1Academy for Engineering & Technology, Fudan University, Shanghai 200433, China; dszhu20@fudan.edu.cn (D.Z.); xdchen20@fudan.edu.cn (X.C.); 2School of Information Science and Technology, Fudan University, Shanghai 200433, China; wangqiuyang21@m.fudan.edu.cn (Q.W.); jbwu21@m.fudan.edu.cn (J.W.); liushuo21@m.fudan.edu.cn (S.L.)

**Keywords:** deep learning, medical image processing, medical diagnosis, diabetic retinopathy, contrastive learning, fundus image

## Abstract

Many researchers have realized the intelligent medical diagnosis of diabetic retinopathy (DR) from fundus images by using deep learning methods, including supervised contrastive learning (SupCon). However, although SupCon brings label information into the calculation of contrastive learning, it does not distinguish between augmented positives and same-label positives. As a result, we propose the concept of Angular Margin and incorporate it into SupCon to address this issue. To demonstrate the effectiveness of our strategy, we tested it on two datasets for the detection and grading of DR. To align with previous work, Accuracy, Precision, Recall, F1, and AUC were selected as evaluation metrics. Moreover, we also chose alignment and uniformity to verify the effect of representation learning and UMAP (Uniform Manifold Approximation and Projection) to visualize fundus image embeddings. In summary, DR detection achieved state-of-the-art results across all metrics, with Accuracy = 98.91, Precision = 98.93, Recall = 98.90, F1 = 98.91, and AUC = 99.80. The grading also attained state-of-the-art results in terms of Accuracy and AUC, which were 85.61 and 93.97, respectively. The experimental results demonstrate that Angular Margin is an excellent intelligent medical diagnostic algorithm, performing well in both DR detection and grading tasks.

## 1. Introduction

According to the International Diabetes Federation’s (IDF) *Diabetes Atlas, 10th edition*, 537 million adults over the age of 20 have diabetes [1]. Diabetes-related eye disease affects at least one-third of all diabetics, of which diabetic retinopathy (DR) is the most common [2]. DR occurs when excessive blood sugar levels damage the capillaries and blood vessels in the retina [3]. If the eye is treated early in the disease’s progression, it can significantly reduce the chance of blindness by up to 98% [4,5]. Therefore, regular ophthalmic screening and timely diagnoses for diabetic patients are necessary [6]. An early clinical diagnosis mainly relies on a fundus examination. Color Fundus Photography (CFP) is a rapid, noninvasive, well-tolerated, and widely used imaging technique to assess the grading of DR [7]. However, the manual CFP-based diagnosis of DR requires the expertise and efforts of specialized ophthalmologists. Especially in underdeveloped areas with large populations, it is difficult to have sufficient medical resources to meet local needs [8]. These conditions highlight the need for the development of intelligent computer-aided diagnosis systems.

With improvements in computational resources and capabilities, deep-learning-related technologies are widely used. A large number of intelligent medical diagnostic technologies are constantly being proposed. In particular, the popular contrastive learning technology has left a deep impression on academia in the past two years [9,10]. Supervised contrastive learning (SupCon) currently achieves great performance in the detection and grading of DR [11]. SupCon is essentially an embedding representation learning method. It makes up for the problem that the cross-entropy loss function is sensitive to noisy data and hyperparameter changes [12]. However, there is certainly an opportunity for development with this method. SupCon adds the supervision signal of the labels to the loss function and treats the same-label data and the augmented data as same-level positives in a coarse-grained manner. But actually, when taking an image as an anchor, the augmented data should be closer to the anchor than the same-label data in the embedding space. For this consideration, our research proposes Angular Margin on the basis of SupCon, which enhances the representation embedding of the fundus image by setting different margins (Figure 1). The experimental results reveal that the Angular Margin approach achieves remarkable performance on the APTOS 2019 dataset for DR detection, with Accuracy = 98.91, Precision = 98.93, Recall = 98.90, F1 = 98.91, and AUC = 99.80. Moreover, for DR grading on the same dataset, it achieves Accuracy = 85.61, Precision = 78.44, Recall = 68.32, F1 = 71.67, and AUC = 93.97, which are significantly superior to the results of other methods. Furthermore, when directly comparing these performance metrics with the SupCon method on the Messidor-2 dataset, the Angular Margin approach demonstrates a substantial improvement across all aforementioned metrics.

For a better theoretical understanding of the role of Angular Margin, we introduced alignment and uniformity [13] into the training process of SupCon. In the embedding space, alignment measures the semantic correlation between positive samples, and uniformity measures the distribution of all samples. The comparison experiment shows that the addition of Angular Margin can improve SupCon’s performance in the above two analysis metrics. Moreover, we also conducted a visual analysis on both SupCon and Angular Margin to strengthen the experimental conclusions. UMAP (Uniform Manifold Approximation and Projection) [14] is a advanced manifold learning technique for dimensionality reduction. It is competitive with t-SNE in terms of visualization quality and preserves more global structure with an excellent runtime performance. In this work, we used UMAP to reduce the dimensionality of the image embeddings to a two-dimensional plane for visualization and compared Angular Margin with the original SupCon.

To sum up, the main innovations of this study are as follows:(1)We propose a novel intelligent medical diagnostic method called “Angular Margin”, which, within the framework of SupCon, enhances the Accuracy of fundus image representation by additionally introducing Angular Margin. This method achieves state-of-the-art performance in both DR detection and grading tasks.(2)Alignment and uniformity are innovatively introduced as evaluation metrics to assess the representational capacity of CFP embeddings.(3)UMAP is introduced to project the embeddings of image representations onto a two-dimensional plane for visualization, enhancing the observation of fundus embeddings.

## 2. Related Work

The development of SupCon is inseparable from self-supervised representation learning, metric distance learning, and supervised learning. Excellent self-supervised representation learning algorithms [9,15] have greatly contributed to academia by enabling noise contrastive estimation and N-pair losses [16,17]. Generally, the loss function is connected to the network’s last layer. But at inference time, it is more likely to use the embeddings of the previous layer for downstream tasks. The loss based on metric distance learning with triplets is similar to contrastive learning [18,19]. The positive and negative logarithms of each data anchor are what distinguishes triplet loss from contrastive loss. Triplet loss uses only one positive and negative pair per anchor [20,21,22]. However, contrastive loss has only one positive pair, but the number of negative pairs is very large. This makes hard-negative mining unnecessary in contrastive loss. Later, inspired by [23,24,25], SupCon [10] was derived, which normalizes embeddings and replaces Euclidean distances with inner products. At the same time, the model capability is further enhanced by using various data augmentation methods. Similar to self-supervision, in the classification task, SupCon adopts a two-stage training method (contrastive learning and cross-entropy) and discards the contrastive head in the second stage.

In recent years, in the field of fundus diagnosis, a large number of deep-learning-related works have also emerged. DR binary and multiclassification works by [26,27,28] were based on the traditional cross-entropy loss function with convolutional neural networks. Quite a few attention mechanism networks [29,30,31,32] and ensemble algorithms [33,34] have also been applied to DR diagnosis. Moreover, several recent studies [35,36,37] have employed more advanced CNN models or introduced additional auxiliary models to enhance the grading of DR. And some approaches [38,39] have utilized a multi-stage training methodology to progressively improve the model’s performance. All these methods have involved good innovations in the structure of the model and achieved the state-of-the-art results at that time. However, the network structures in most of these works are complex and large. IsIam et al. [11] recently brought DR detection and classification to a new level without adding additional modules by introducing SupCon. This shows the potential of SupCon for the application of fundus imaging. In addition, Huang et al. [40] proposed a self-supervised model for lesion-based contrastive learning for DR grading, and it performs well on the EyePACS dataset. Cai et al. [41] used optical coherence tomography (OCT) and CFP for multi-modality supervised contrastive learning to diagnose glaucoma. In two papers, Cheng et al. [42,43] used two contrastive learning methods to improve the imaging quality of fundus images.

## 3. Methodology

This section mainly introduces the datasets used in the experiments and the details of our methods. The innovative methods proposed in this section have yielded excellent results, which can be reviewed in Section 4.

### 3.1. Datasets

Currently, there are many datasets for DR detection and grading, but to align with previous methods, we chose APTOS 2019 (download link: https://www.kaggle.com/c/aptos2019-blindness-detection (accessed on 5 March 2022)) and Messidor-2 [44,45].

APTOS 2019 was provided by Aravind Eye Hospital in India and includes five grades to identify and assess the severity of DR: no DR, mild, moderate, severe, and proliferative diabetic retinopathy (PDR). The number of training samples in the dataset is 3662, and the testing part consists of 1928 samples. The image sizes are 2416 × 1736, 819 × 614, and 3216 × 2136. Although test samples are available, their labels are not public and are only available for online submission for testing. Therefore, our work does not consider the test part.

The Messidor-2 dataset consists of more than 1700 images of various sizes, such as 1440 × 960, 2240 × 1488, and 2340 × 1536, and is used for DR grading. The partial dataset (Messidor-Original) was kindly provided by the partners of the Messidor project, and the rest (Messidor-Extension) was provided by the University Hospital of Brest, France.

We further split the above dataset into training (70%), validation (15%), and testing (15%) sets for DR detection and severity grading. The data have the problem of being highly imbalanced, so they need to be sampled evenly when splitting. Multiclass classification is performed for the five-stage grading of DR severity, and binary classification is carried out for DR detection. Table 1 summarizes the specifics of the dataset.

### 3.2. Contrastive Learning Framework

SupCon is an advanced deep learning method that aims to improve the discriminative power of learned representations. By leveraging labeled data, SupCon uses a contrastive loss function to encourage similar representations for data instances from the same class while pushing apart representations from different classes. This approach facilitates the extraction of meaningful and discriminative features for tasks such as classification and clustering. SupCon maximizes the agreement between augmented views of the same sample and minimizes the agreement between different samples, promoting the learning of informative representations. It has shown success in various domains and offers a promising direction for representation learning by utilizing unlabeled data.

Our framework is primarily composed of a data augmentation module (Aug(·)), an encoder network (Enc(·)), a projection layer (Proj(·)), and a classifier (Class(·)). The pipeline of the whole framework is shown in Figure 2 in detail.

For each input data *x*, we randomly generate two augmentations $\hat{x}=Aug\left(x\right)$. Each augmentation represents a different view of the image and contains some subset information of the original data. The image augmentation methods operate on the principle of making simple direct modifications to an image, such as cropping, rotating, adjusting the color, and changing the image contrast. These techniques aim to generate another image that is visually similar to the original one. We used RandomResizedCrop, RandomHorizontalFlip, and RandomVerticalFlip (the augmentation methods come from the *transforms* module of *torchvision*; official link: https://pytorch.org/vision/stable/transforms.html (accessed on 10 May 2022)) to construct the data augmentation module.

The encoder is a convolutional neural network (CNN)-structured backbone that maps *x* to a vector $r=Enc\left(x\right)\in R^{D_{E}}$ as the representation embedding. To obtain a pair of representation embeddings, two augmentations are fed into the same encoder. We follow the works of [10,13,19] to normalize *r* to the unit hypersphere in $R^{D_{E}}$. The normalization utilized here involves scaling each component of the embedding to fall within the range between 0 and 1. In this work, ResNet [46] was chosen as the encoder. ResNet is a convolutional neural network architecture that addresses the degradation problem in deep networks. Proposed in 2015, it introduces residual learning and utilizes residual blocks with skip connections to learn residual mappings. ResNet models, such as ResNet-18 and ResNet-50, have achieved impressive performance in computer vision tasks. Its widespread adoption has made it a benchmark in the field and influenced subsequent network designs.

The projection transforms *r* to the vector $z=Proj\left(r\right)\in R^{D_{P}}$. Proj(·) is instantiated as a 2-layer perceptron [47] with a 2048 hidden layer size and a 128 output layer size. Previous studies [10,11] also normalized the Proj(·) result to lie on the unit hypersphere so that the inner product could be used to measure distances in a projective space. At the end of training, we drop Proj(·) so that the contrastive learning framework has the same number of parameters as the cross-entropy model.

The classifier’s objective is to compute the cross-entropy loss. Because our main idea in this research is the classification of DR, even with Angular Margin loss, it is still necessary to train a classifier for classification.

### 3.3. Angular Margin Loss Function

To better demonstrate the Angular Margin loss, we need to introduce a family of contrastive learning losses. For a set of N randomly sampled data, ${\left\{x_{k},y_{k}\right\}}_{k=1,...,N}$, so there are 2N pairs of corresponding batches actually used for training after augmentation, ${\left\{{\hat{x}}_{l},{\hat{y}}_{l},\right\}}_{l=1,...,2N}$. The collection of N data is referred to as a “batch”, and the set of 2N augmented data is referred to as a “double-viewed batch”. This setting is common to all contrastive learning losses.

Within a double-viewed batch, i∈I≡{1,...,2N} represents the index of any data, and j(i) is the corresponding augmented data. In the self-supervised paradigm, the loss usually takes the following form:(1)$$L^{self}=\sum_{i\in I}L_{i}^{self}=-\sum_{i\in I}\log\frac{\exp(z_{i}\cdot z_{j(i)}/\tau )}{\sum_{a\in A(i)}\exp\left(z_{i}\cdot z_{a}/\tau \right)}$$

In the formula, $z_{l}=Proj\left(Enc\left({\hat{x}}_{l}\right)\right)\in R^{D_{P}}$, the symbol · is the vector dot product, $\tau \in R^{+}$ represents the temperature parameter, and A(i)≡I∖{i}. The index *i* can be regarded as an anchor, j(i) is the corresponding positive, and all other samples in the double-viewed batch are regarded as negatives. So, in the self-supervised paradigm [9,20], each sample has 1 positive pair and 2N−2 negative pairs. However, the label information of the data is not taken into account, and the coarse-grained thinking that there is only one positive pair in a double-view batch is also problematic.

To address this issue, Khosla et al. [10] proposed a supervised paradigm. Equation (Equation 2) extends the contrastive learning loss to multiple positive pairs by introducing label information:(2)$$L^{sup}=\sum_{i\in I}L_{i}^{sup}=\sum_{i\in I}\frac{-1}{\left|P(i)\right|}\sum_{p\in P(i)}\log\frac{\exp(z_{i}\cdot z_{p}/\tau )}{\sum_{a\in A(i)}\exp\left(z_{i}\cdot z_{a}/\tau \right)}$$

P(i) is the set of positives in the double-viewed batch. In this way, in the framework of contrastive learning, an anchor can have multiple positives and negatives. But at the same time, multiple positive pairs also expose the potential problem that not every positive is at the same similarity level for the anchor. As shown in Figure 1, the augmented data from anchor data should naturally be closer to the anchor than the same-label data. Therefore, we propose the Angular Margin loss to increase the degree of discrimination between positives.

To add the angle calculation to the loss, we need to replace the previous vector dot product with cosine similarity $\frac{z_{i}\cdot z_{p}}{∥z_{i}∥*∥z_{p}∥}$. Let us assume that $\theta_{i,p}$ has the following form:(3)$$\theta_{i,p}=\arccos\left(\frac{z_{i}\cdot z_{p}}{∥z_{i}∥*∥z_{p}∥}\right)$$

Through Equation (Equation 3), we can convert the dot product of vectors into an angle representation so that we can add margins to increase the discrimination between positives.
(4)$$\begin{array}{cc} L^{a-m} & =\sum_{i\in I}L_{i}^{a-m}\\ L_{i}^{a-m} & =\sum_{i\in I}\frac{-1}{\left|P(i)\right|}\sum_{p\in P(i)}\log\frac{\exp(\cos\left(\theta_{i,p}+I_{U}^{p}m_{u}+I_{V}^{p}m_{v}\right)/\tau )}{\sum_{a\in A(i)}\exp\left(\cos\left(\theta_{i,a}+I_{U}^{a}m_{u}+I_{V}^{a}m_{v}\right)/\tau \right)} \end{array}$$

*U* indicates the augmented positives in the double-viewed batch, and *V* indicates the same-label positives; both belong to P(i) (|U|+|V|=|P(i)|). I∈{0,1} is the indicator function, which is equal to 1 if and only if the superscript belongs to the subscript. To strengthen the discriminative power, we add $m_{u}$ and $m_{v}$, corresponding to augmented positives and same-label positives, respectively (Figure 3). Their values are expressed in radians.

The final loss function (Equation (Equation 6)) is obtained by combining Equation (Equation 4) and the cross-entropy loss function (Equation (Equation 5)) through the hyperparameter λ.
(5)$$\begin{array}{c} L^{ce}=\left\{\begin{array}{ccc} \; & -\sum_{i=1}^{k}y_{i}\log\left(p_{i}\right), & multiclass\\ \; & -[y\log(p)+(1-y)\log(1-p)], & binary \end{array}\right. \end{array}$$
(6)$$L=L^{ce}+\lambda L^{a-m}$$

### 3.4. Alignment and Uniformity

Wang et al. [13] discovered two important properties related to contrastive learning, alignment and uniformity. Given the normalized distribution of positive pairs $p_{pos}$, alignment computes the expectation of the distance between embeddings of the pairs:(7)$$L_{align}≜\underset{\left(x,y\right)∼p_{pos}}{E}{∥f\left(x\right)-f\left(y\right)∥}^{2}$$

At the same time, given a dataset $p_{data}$, uniformity measures the degree of uniform distribution of the embeddings.
(8)$$L_{uniform}≜\log\underset{\left(x,y\right)\overset{i.i.d.}{∼}p_{data}}{E}e^{{-2∥f\left(x\right)-f\left(y\right)∥}^{2}}$$

For the above two metrics, the smaller the value, the better the learning of the embedding representation. In this work, we use them to demonstrate that our method is better than the standard SupCon.

## 4. Experiments and Results

### 4.1. Experimental Protocol

To pick the best model for the test set, we trained the network for 20 epochs and tested it on the validation set at the conclusion of each epoch. The size of the input image was 224 × 224. The augmentation methods were RandomResizedCrop (scale = (0.8, 1.0)), RandomHorizontalFlip (*p* = 0.5), and RandomVerticalFlip (*p* = 0.5). The batch size was 64, the learning rate was 2 × 10${}^{-4}$, and the optimizer used was Adam. These hyperparameters were obtained through empirical means. Among other hyperparameters, τ was 0.05, $m_{u}$ was 0.2, $m_{v}$ was 0.1, and λ was 1. The device we used was NVIDIA A4000 GPU.

### 4.2. Metrics for Classification

There are five evaluation metrics, namely, Accuracy, Precision, Recall, F1-score, and AUC. Accuracy refers to the proportion of correctly predicted samples to total samples. The F1-score (Equation (Equation 11)) is a combination of Precision and Recall. It will perform well only if both Precision and Recall scores are high. *Precision* (Equation (Equation 9)) and *Recall* (Equation (Equation 10)) consist of True Positives (*TP*), False Positives (*FP*), and False Negatives (*FN*). AUC (Area Under the Curve) is defined as the area contained by the ROC curve and the coordinate axis. The values of the above metrics are all between 0 and 1. For convenience, we express them in percentiles (%).
(9)$$Precision=\frac{TP}{TP+FP}$$
(10)$$Recall=\frac{TP}{TP+FN}$$
(11)$$F1=\frac{2\times Precision\times Recall}{Precision+Recall}$$

### 4.3. Results

We compared our approach with state-of-the-art methods on the APTOS 2019 dataset (Table 2). The APTOS 2019 dataset consists of a large number of fundus images, and significant research efforts have been devoted to training and testing on this dataset. Therefore, the comparative experiments presented in this study were conducted on the APTOS 2019 dataset. For the binary classification task, our method obtains state-of-the-art performance across all metrics, improving by more than 0.5% over the already achieved 98% baseline. But in multiclassification tasks, our method only achieves state-of-the-art results for Accuracy and AUC. A possible reason may be that Bodapati et al. [26] used more data (their training set accounts for 80% of APTOS 2019, whereas ours is only 70%) with their method and introduced more fully connected network layers (additional FC1 and FC2).

To assess the effectiveness of Angular Margin in its entirety, the experiment also compared differences in alignment and uniformity between SupCon and Angular Margin. The lower the values of the two metrics, the better the representation effect. We took the first 1500 steps of the training process and calculated the alignment and uniformity of the current batch every 10 steps. Figure 4 and Figure 5 depict alignment and uniformity, respectively. In Figure 4, all lines gradually decrease, indicating that the alignment gradually decreases with training until it becomes stable. After 200 to 600 steps, the line of Angular Margin is obviously located below that of SupCon, showing that Angular Margin can bring the positives closer. However, in Figure 5, the value of uniformity either increases gradually or oscillates repeatedly, failing to drop to lower values like alignment does. We estimate that there may be a lot of data with the same labels in a batch because the number of classes is too small (only two classes for detection and five classes for grading). Therefore, after optimizing the alignment, it is difficult for the embeddings to maintain a standard uniform distribution. Even so, from the performance in the figure, the effect of Angular Margin is still better than that of SupCon.

To understand the advantages and characteristics of Angular Margin more specifically, we used UMAP [14] for visualization. UMAP is a dimensionality reduction technique used to visualize high-dimensional data in a lower-dimensional space. It preserves both the local and global structures of the data by constructing a topological representation. UMAP is computationally efficient, can handle large datasets, and does not require labeled data. It has gained popularity in machine learning, bioinformatics, and data visualization for capturing complex data structures and revealing patterns. UMAP concatenates 2048-dimensional representation embeddings and projects them onto a 2-dimensional plane. Figure 6 shows the results of SupCon and Angular Margin on the APTOS 2019 and Messidor-2 test sets. The binary classification task is relatively simple, so we chose the multiclassification task for visual representation. This allows a clearer gap to be seen in the visualization. Judging from the results in Figure 6, since the classification results of APTOS 2019 are better than those of Messidor-2, the discrimination shown in Figure 6a,b is better than that in Figure 6c,d. On the same dataset, the distinguishing ability of the representation of Angular Margin on the two-dimensional projection is stronger than that of SupCon. Figure 6a shows that SupCon is able to clearly divide the embeddings into normal and DR, but the grading of DR is still vague. Figure 6b introduces Angular Margin, which can more clearly distinguish between severe (3.0) and PDR (4.0). Similarly, although the clustering of embeddings in Figure 6c,d is not very clear, separate independent clusters are more easily discerned in Figure 6d than in Figure 6c.

### 4.4. Ablation Study

Ablation experiments based on the original SupCon and Angular Margin were conducted on two datasets, APTOS 2019 and Messidor-2. Table 3 records the results of experiments on APTOS 2019. It can be seen that each method works well in the task of binary classification, but Angular Margin is still one percentage point higher on average. The advantage of Angular Margin is amplified in the multiclassification task, with an average improvement of more than 4 percentage points. Table 4 lists the results of Messidor-2 and also depicts a similar situation. There is an average improvement of more than 4 percentage points in binary classification and multiclassification. The results in the two tables reflect that Angular Margin can indeed make a big improvement compared to SupCon.

There are two particularly important hyperparameters in the Angular Margin loss function, $m_{u}$ and $m_{v}$. Because these two angle values are directly related to the relative angle relationship between positives and anchor data, they will affect the distinguishing effect of representation embedding. For this reason, this study deliberately conducted detailed ablation experiments to explore its specific impact on the results. The experiments were compared on the basis of multiclassification tasks on the Messidor-2 and APTOS 2019 datasets. In Figure 7, each polyline has a certain value of $m_{v}$, and there are five values in total: 0.02, 0.05, 0.1, 0.2, and 0.4. The vertical axis is the value of AUC on the test set. Compared with other evaluation indicators, AUC is more balanced and comprehensive. The horizontal axis represents the value of $m_{u}$, which is a multiple of $m_{v}$, because it is easier to find a reliable rule than arbitrary values. The minimum multiple of $m_{u}$ is 1.0, which represents the special case where $m_{u}$ and $m_{v}$ have the same value. Angular Margin will degenerate to a SupCon-like form. The highest multiple of $m_{u}$ is 3.0, because the angle value calculated by cosine similarity has a range, and too large an angle value will lead to more errors. In fact, the results in Figure 7 also reflect this. The five polylines at the position of 1.0 are relatively close, because they all have a structure similar to SupCon. The position at 3.0 is the worst performance of each polyline, and some are even lower than the results of SupCon, because the excessive $m_{u}$ has led to errors in the loss calculation. From the perspective of the two subgraphs, $m_{v}$ is 0.1, and $m_{u}$ is twice as high as $m_{v}$, which is the best setting.

The augmentation module of contrastive learning has always been a key point and has a very large impact on the results. In this study, six commonly used image augmentation methods were selected, and the relationship between them is described with a heatmap. The six augmentation methods are RandomResizedCrop, RandomHorizontalFlip, RandomGrayscale, RandomVerticalFlip, ColorJitter, and RandomRotation. In Figure 8, the values in the heatmap are the results for AUC on the test set. Because the horizontal and vertical axes on the main diagonal refer to the same method, the augmentation module contains only one augmentation method. The parts other than the main diagonal refer to two augmentation methods. From the results in the figure, it can be seen that RandomResizedCrop, RandomHorizontalFlip, and RandomVerticalFlip have the best compatibility. They contribute the most to the improvement in results.

## 5. Conclusions

Inspired by SupCon, this paper proposes an improved method called Angular Margin based on it. To test the effect of this training strategy, in this study, the theory and formula were first deduced, and then detailed tests were conducted on APTOS 2019 and Messidor-2. Angular Margin could better distinguish positives with the anchor and achieved state-of-the-art results according to multiple classification metrics by combining the cross-entropy loss function. Alignment and uniformity were also introduced in the experiment, and the embedding effect of Angular Margin was found to be better than that of ordinary SupCon through comparison. In addition to these numerical comparisons, we also include visualizations of representational embeddings. A visualization of embeddings projected to 2D space by UMAP clearly shows that the results for Angular Margin are more spatially clustered. Finally, various ablation experiments were performed to investigate the role of Angular Margin loss, the effect of the angular value, and the difference between different augmentation methods as comprehensively as possible. To summarize, this paper confirms that Angular Margin performs better than SupCon in the intelligent medical diagnosis of DR.

The results of the experiments indicate that the proposed method is relatively successful, but there are still some areas worthy of improvement. First of all, the improvement scheme that we propose must not be optimal. Determining how to better use the contrastive learning framework and apply it to actual scenarios is a very valuable direction. Second, the presented experiments were all conducted on public datasets. In the future, we look forward to actively seeking cooperation with hospitals, hoping to obtain more data to conduct some clinical experiments. Finally, Angular Margin is a mechanism for representing images in general. We believe that it has the potential to be applied to other elements of ophthalmological medical diagnostics, as well as the diagnosis of other human tissues. This part of the work can be left to other researchers for further refinement. 

## Figures and Tables

**Figure 1 diagnostics-13-02389-f001:**
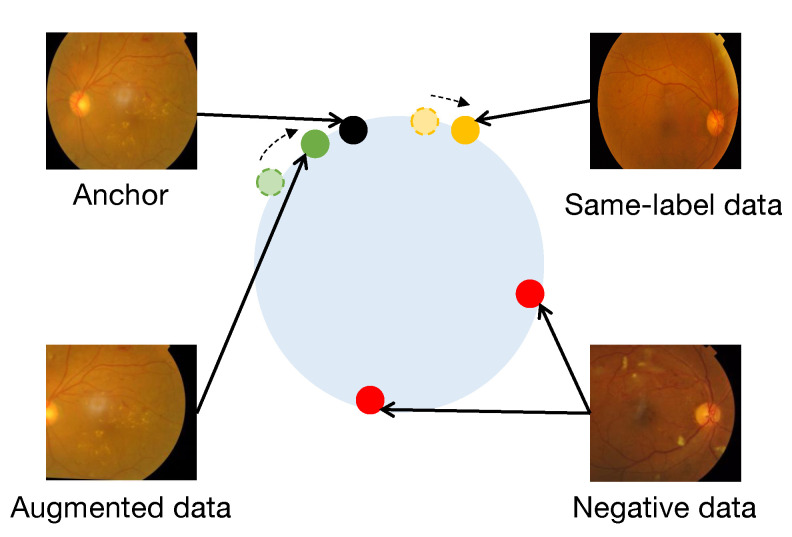
The figure shows the role of Angular Margin in SupCon from fundus images. In the traditional SupCon, both the augmented data and the same-label data in the batch belong to positive samples. But in practice, it is reasonable that the augmented data should be closer to the anchor in the embedding space, because they are from the same image. The addition of Angular Margin can solve this problem well, making the representation of fundus images more accurate in the embedding space.

**Figure 2 diagnostics-13-02389-f002:**
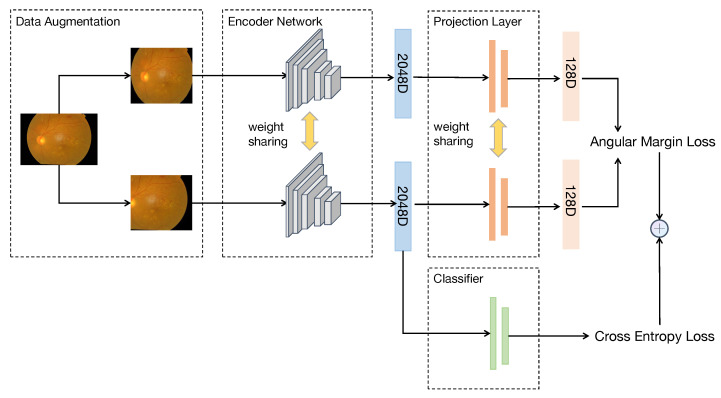
The whole model training process is from left to right. A fundus image can generate two sub-images with different views after passing through the data augmentation module. Afterward, through the encoder network, sub-images can be turned into embeddings. Finally, the Angular Margin loss and cross-entropy loss are calculated through the output of the projection layer and the classifier, respectively. The two losses are combined to form a completed training framework.

**Figure 3 diagnostics-13-02389-f003:**
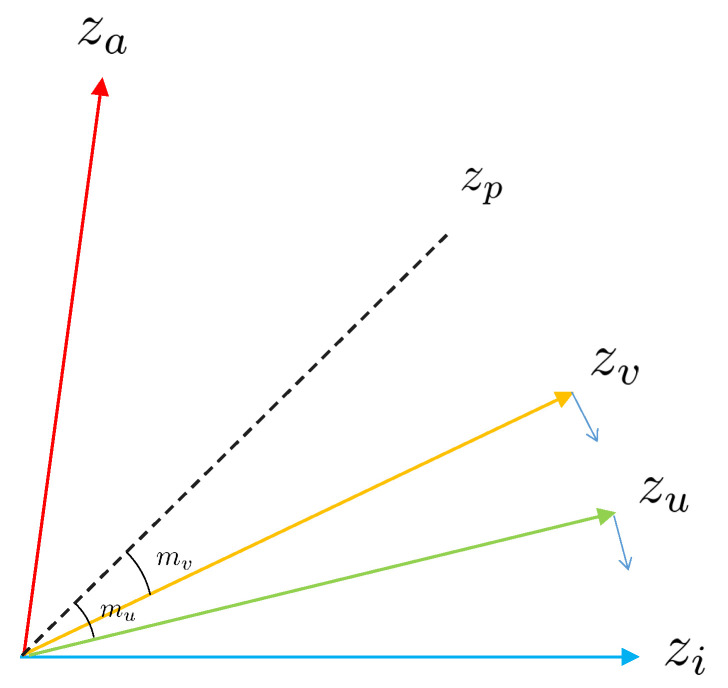
The figure shows the angle relation of representation embedding. We assume that $z_{i}$ is an anchor, $z_{a}$ is the corresponding negative, and $z_{p}$ is the positive. After introducing Angular Margin, $z_{u}$ corresponds to the augmented positive, and $z_{v}$ corresponds to the same-label positive. Because $m_{u}$ and $m_{v}$ are set to different sizes, $z_{u}$ is pushed closer to the anchor than $z_{v}$.

**Figure 4 diagnostics-13-02389-f004:**
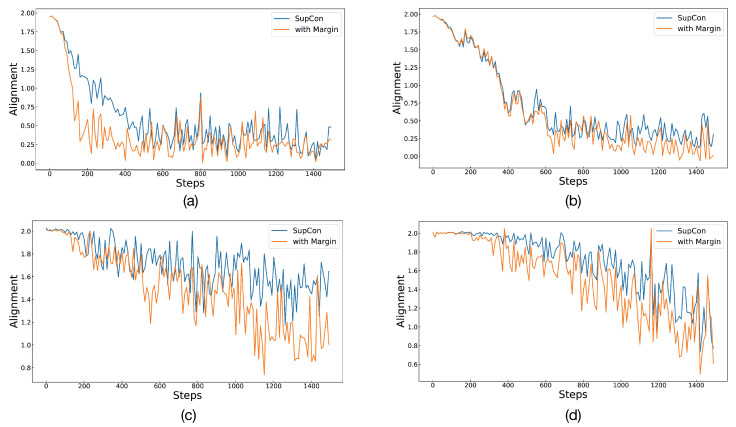
The figure shows the situation of alignment for SupCon and Angular Margin. Graphs in (**a**,**b**) represent binary classification and multiclassification on APTOS 2019. Similarly, (**c**,**d**) represent the two classification tasks on Messidor-2.

**Figure 5 diagnostics-13-02389-f005:**
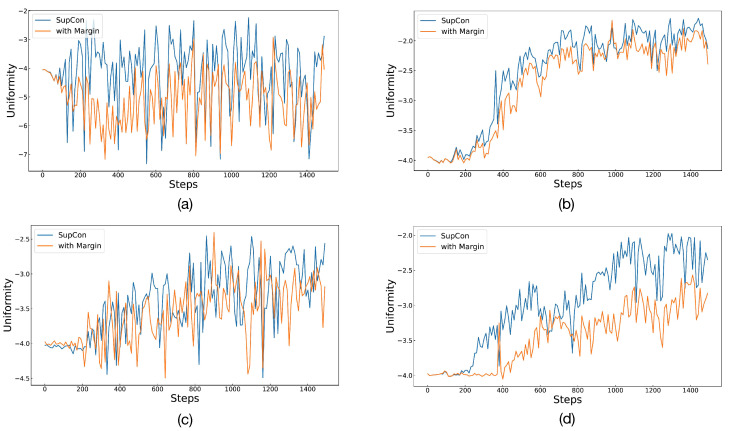
The figure shows the situation of uniformity for SupCon and Angular Margin. Graphs in (**a**,**b**) represent binary classification and multiclassification on APTOS 2019. Similarly, (**c**,**d**) represent the two classification tasks on Messidor-2.

**Figure 6 diagnostics-13-02389-f006:**
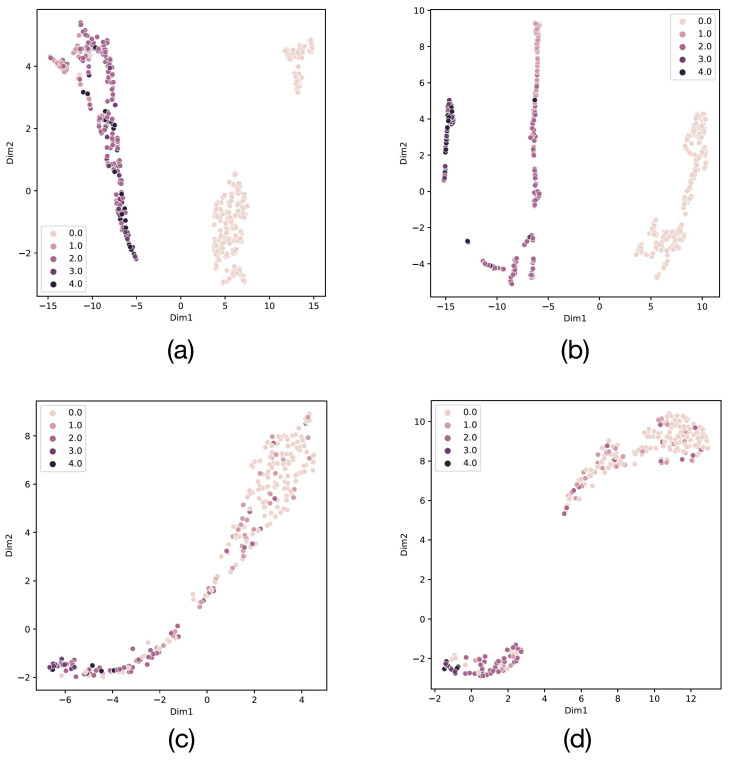
(**a**,**b**) The results of SupCon and Angular Margin on APTOS 2019. (**c**,**d**) The results of SupCon and Angular Margin on Messidor-2. 0.0 to 4.0, which indicate normal to PDR, and the larger the number, the darker the color, and the more serious the DR.

**Figure 7 diagnostics-13-02389-f007:**
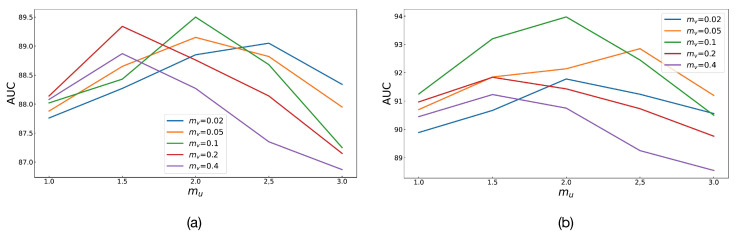
Comparison of ablation experiments with angle values, $m_{u}$ and $m_{v}$. (**a**) The case of the Messidor-2 dataset and (**b**) the case of the APTOS 2019 dataset.

**Figure 8 diagnostics-13-02389-f008:**
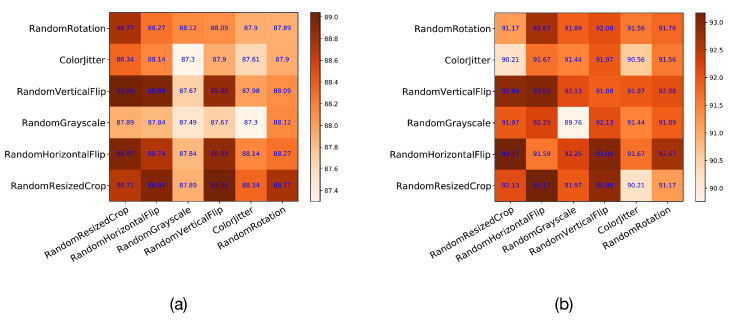
The relationship heatmap of 6 commonly used augmentation modules. (**a**) The results of Messidor-2 and (**b**) the results of APTOS 2019. The effects of pairwise combinations of augmentation methods on the experimental results are described in the form of diagonal matrices.

**Table 1 diagnostics-13-02389-t001:** The table covers each component of APTOS 2019 and Messidor-2 after division. In the multiclass part, there are serious data imbalances in both datasets.

Classification	DR Stage	Training Part		Validation Part		Testing Part	
		APTOS 2019	Messidor-2	APTOS 2019	Messidor-2	APTOS 2019	Messidor-2
Multiclass	No DR	1263	711	271	153	271	153
	Mild	259	189	56	40	55	41
	Moderate	699	243	150	52	150	52
	Severe	135	52	29	12	29	11
	PDR	207	25	44	5	44	5
Binary	No DR	1263	711	271	153	271	153
	DR	1300	509	279	109	278	109

**Table 2 diagnostics-13-02389-t002:** Our method compared with previous state-of-the-art methods in binary and multiclassification tasks on APTOS 2019. The data in the table are from the results in the original papers of the respective authors. Boldface type indicates the best-performing result among the metrics.

Method	Accuracy	Precision	Recall	F1	AUC
*Binary class*					
Bodapati et al. [48]	97.34	97.00	97.00	97.00	/
Bodapati et al. [26]	97.82	98.00	98.00	98.00	98.00
Islam et al. [11]	98.36	98.37	98.36	98.37	98.50
This study	**98.91**	**98.93**	**98.90**	**98.91**	**99.80**
*Multiclass*					
Dondeti et al. [27]	77.90	76.00	77.00	75.00	/
Bodapati et al. [48]	79.73	78.00	78.00	78.00	/
Kassani et al. [49]	79.59	/	82.35	/	/
Bodapati et al. [26]	82.54	**82.00**	**83.00**	**82.00**	79.00
Kobat et al. [37]	84.90	74.32	68.53	70.75	/
Islam et al. [11]	84.36	70.51	73.84	70.49	93.83
This study	**85.61**	78.44	68.32	71.61	**93.97**

**Table 3 diagnostics-13-02389-t003:** Experimental results of SupCon and Angular Margin (A-M) on the APTOS 2019 dataset.

Metrics	Binary Class		Multiclass	
	SupCon	with A-M	SupCon	with A-M
Accuracy	96.90	98.91	81.09	85.61
Precision	96.94	98.93	74.58	78.44
Recall	96.92	98.90	61.72	68.32
F1	96.90	98.91	67.26	71.61
AUC	99.58	99.80	89.75	93.97

**Table 4 diagnostics-13-02389-t004:** Experimental results of SupCon and Angular Margin (A-M) on the Messidor-2 dataset.

Metrics	Binary Class		Multiclass	
	SupCon	with A-M	SupCon	with A-M
Accuracy	75.95	80.53	67.94	72.14
Precision	75.89	81.35	51.87	70.09
Recall	76.64	78.45	50.37	56.40
F1	75.77	79.17	50.41	59.46
AUC	87.01	87.31	87.23	89.50

## Data Availability

Publicly available datasets were analyzed in this study. These data can be found here: https://www.kaggle.com/c/aptos2019-blindness-detection (accessed on 5 March 2022).

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
