# Peer review of "Supervised Contrastive Learning with Angular Margin for the Detection and Grading of Diabetic Retinopathy"

_diagnostics, 2023, doi:10.3390/diagnostics13142389_

Round 1
Reviewer 1 Report
This is a technically challenging area for most ophthalmologists who lack the engineering background to fully appreciate the authors' contribution. In order to achieve greater recognition of their work, and enhance value to the readers, several paragraphs of information related to the basic concepts of SupCon and Angular Margin must be added to the paper. In addition, simplification and use of non-mathematical, non-engineering terminology in paragraphs prior to the technical presentations would be important. This additional effort will broaden the readership and increase the citations of the paper.
Reviewer 2 Report
· “the” is used twice in line 134.
· There is a typo used as "anglur" in the Figure 3 title.
· Corrections are required as to the spelling language.
· The UMAP expression in the summary should be introduced.
· In the summary part, the results obtained from the study should be briefly mentioned.
· At the end of the introduction, it will be useful to give brief information about the parts and content of the study.
· In the introduction, the results of the study should be mentioned.
· In the introduction, the contributions of the study to the literature and the aspects that differ from other studies should be detailed.
· It will be useful to explain the modules used in data augmentation in detail.
· Reference is given for normalization at the encoder stage, but it is useful to briefly explain how normalization was made.
· The ResNet encoder used must be introduced.
· How were the parameters determined for the training determined?
· It would be more appropriate to use "This Study" instead of "ours" in Table 2.
· The comparison in Table 2 was made according to the results obtained from which data set?
· Placing figures and tables in the relevant section will make the study more readable.
· It is useful to mention the achievements obtained in the study in the solution section.
· More recent references on the subject of study can be consulted. There are studies done in recent years.
Minor editing of the English language required
